# Antimicrobial Susceptibility Determination of Less Frequently Isolated *Legionella* Species by Broth and Agar Dilution

**DOI:** 10.3390/antibiotics14111165

**Published:** 2025-11-17

**Authors:** Caitlin Farley, Amy Price, Max Sewell, Rachael Barton, Edward A. R. Portal, Ian Boostrom, Jessica Day, Baharak Afshar, Victoria J. Chalker, Owen B. Spiller

**Affiliations:** 1Department of Medical Microbiology, Division of Infection and Immunity, Cardiff University, University Hospital of Wales, Cardiff CF14 4XN, UK; farleyc@cardiff.ac.uk (C.F.); pricea39@cardiff.ac.uk (A.P.); sewellm1@cardiff.ac.uk (M.S.); bartonrm@cardiff.ac.uk (R.B.); edward.portal@biology.ox.ac.uk (E.A.R.P.); boostrom@cardiff.ac.uk (I.B.); victoria.chalker@nhs.net (V.J.C.); 2Ineos Oxford Institute for Antimicrobial Research, Department of Biology, University of Oxford, Oxford OX1 3RE, UK; 3Respiratory and Vaccine Preventable Bacteria Reference Unit (RVPBRU), United Kingdom Health Security Agency, London NW9 5EQ, UK; jessica.day@ukhsa.gov.uk (J.D.); baharak.afshar@ukhsa.gov.uk (B.A.); 4NHS England, Wellington House, 133-155 Waterloo Rd., London SE1 8UG, UK

**Keywords:** *Legionella*, *Legionella pneumophila*, non-*pneumophila Legionella*, antimicrobial susceptibility testing

## Abstract

**Background/Objectives**: Infections caused by *Legionella* species are primarily associated with *Legionella pneumophila*, but non-*pneumophila* species are increasingly implicated in human disease. Despite this, antimicrobial susceptibility testing (AST) data for non-*pneumophila* species remain scarce, and standardised testing protocols or resistance thresholds have not been established. This study aimed to address this gap by evaluating and comparing AST performance for non-*pneumophila Legionella* species relative to *L. pneumophila* using three methodologies. **Methods**: AST was conducted on 89 *Legionella* isolates using LASARUS agar dilution, buffered yeast extract broth microdilution (BYE-BMD), and BCYE-α agar dilution, against ampicillin, azithromycin, chloramphenicol, doxycycline, levofloxacin, and rifampicin. Growth performance and minimum inhibitory concentrations (MICs) were assessed after a 96 h incubation. **Results**: MIC profiles were obtained using LASARUS and BYE-BMD for 53.9% and 93.3% of isolates, respectively. While *L. pneumophila* reached sufficient turbidity in BYE-BMD after a 48 h incubation, non-*pneumophila* species required an extended incubation (72–96 h). Non-*pneumophila* species displayed broader MIC ranges against azithromycin (0.016–1 mg/L) and levofloxacin (0.016–0.25 mg/L), but a narrower rifampicin range (≤0.0005–0.032 mg/L) relative to *L. pneumophila*. *L. longbeachae* exhibited a higher MIC_50_ for rifampicin despite overlapping susceptibility ranges across all species (0.001–0.016 mg/L). **Conclusions**: This study demonstrates species-specific differences in *Legionella* susceptibility and highlights the limitations in extrapolating *L. pneumophila*-based AST data. Azithromycin MICs in non-*pneumophila* species exceeded those of *L. pneumophila*, raising clinical concern. While BYE-BMD was the most effective method for MIC determination, three species required BCYE-α due to poor growth. These findings support developing standardised, species-specific AST protocols and thresholds amid rising macrolide resistance and the increasing detection of non-*pneumophila* infections.

## 1. Introduction

To date, the *Legionella* genus comprises 67 *Legionella* species, 30 of which have been identified as human pathogens [1,2]. *Legionella* are fastidious obligate aerobe Gram-negative bacteria and the causative agent of Legionnaires’ Disease (LD), an atypical form of pneumonia, accounting for ~2–9% of all community acquired pneumonia cases worldwide [1,3]. The dominant cause of LD in Europe and the United States is *L. pneumophila*, accounting for approximately 90% of reported cases [1]. In contrast, in Australia and New Zealand, 60–70% of LD cases are caused by *L. longbeachae* [1]. By comparison, *L. longbeachae* accounts for 1.1% and 1.8% of reported cases in the United Kingdom (2017–2023) and the United States (2018–2019), respectively [4,5]. Examples of *Legionella* species less frequently identified as disease contributors include *L. anisa*, *L. bozemanae*, and *L. micdadei*, with clinical manifestations encompassing Pontiac fever, bacterial endocarditis, skin and soft tissue infections, and septic arthritis [1,6,7,8]. *Legionella* species replicate within macrophages, so effective therapy requires antibiotics that penetrate host cells, such as tetracyclines, macrolides, and fluoroquinolones; agents that cannot enter host cells, such as aminoglycosides and β-lactams, are ineffective [3].

*Legionella* species are found predominantly within natural and man-made water systems, including lakes, reservoirs, and air-conditioning cooling towers, ubiquitously across the globe [9]. *L. longbeachae* is an exception to this, as it is primarily found in soil and the bark of pine trees [6]. Within the environment, *Legionella* can adopt different lifestyles as free-living biofilm-associated bacteria or associated with a host [1]. Association with their natural amoebae host species provides protection, increasing the resistance of *Legionella* to antibiotics, biocides, and acids, and provides an environment for replication and increased transmission via the release of air-borne vesicles [10,11].

*Legionella* is increasingly recognised as an important pathogen due to its disease potential, global significance, and the increasing threat it poses with the expansion of man-made water systems [12,13,14]. An increase in diagnostic techniques, awareness, and a movement towards the active monitoring of water sources has resulted from this, including the implementation of the Drinking Water Directive (DWD) law in Europe [12,14]. The gold standard for the detection and quantification of *Legionella* species, for environmental screening and clinical diagnosis, remains culture-based screening, which takes several days for the detection of most *Legionella* species [9,12]. For environmental screening, non-culture-based methods, including qPCR (ISO 12869), have been accepted for the risk-based monitoring of *Legionella*, as these results can be achieved in a shorter time [12]. Clinically, legionellosis is diagnosed using clinical and/or radiological symptoms and laboratory results, with 90% of cases in Europe being diagnosed with a urine antigen test (UAT) [10]. However, as the UAT is biased towards only detecting antigens for serogroup 1 *L. pneumophila*, microbial culture remains the only detection method for other clinically relevant *Legionella* species [15].

Despite an increase in the monitoring and awareness of *Legionella*, there remains a lack of internationally standardised methodology, susceptibility cut-offs, or control strains for the antimicrobial susceptibility testing (AST) of *Legionella* species [16]. Currently, the minimum inhibitory concentrations (MICs) of *Legionella* species have been determined using broth microdilution (BMD), E-test gradient strips on BCYE (Buffered Charcoal Yeast Extract), BCYE agar dilution, and *Legionella* Antimicrobial Susceptibility and Resistance Universal Screening medium (LASARUS) agar dilution [16,17,18]. However, there is a paucity of investigations that benchmark ASTs for other clinically relevant *Legionella* species against *L. pneumophila*, and the limited comparisons that do exist utilise BCYE, an activated charcoal-containing medium that has been shown to be variable and unreliable for AST [16,19,20].

This study aims to provide a foundation for AST comparisons of non-*pneumophila Legionella* species relative to *L. pneumophila*, with the objective of identifying species-suitable methodologies and associated limitations. As no previous investigation have utilised the BMD method for species-level comparison in this context, we hereby employed a proposed standardised BMD protocol alongside both non-charcoal-containing and charcoal-containing agar dilution methodologies.

## 2. Results

### 2.1. Legionella Species Growth by Method of MIC Determination

AST of 89 *Legionella* isolates was conducted using both the LASARUS agar dilution and buffered yeast extract broth microdilution (BYE-BMD). Using LASARUS, complete MICs were collected for 53.9% (n = 48) of the isolates after 96 h. This included all tested *L. pneumophila* and *L. longbeachae* isolates. After 72 h, BYE-BMD obtained complete MIC profiles for 93.3% (n = 83) of the isolates. All *L. pneumophila* MICs could be obtained after 48 h. Six isolates had MICs obtained with BCYE agar dilution, having failed to obtain complete AST profiles with LASARUS or BMD. Table 1 outlines the methods used to obtain the MICs for each of the 34 *Legionella* species investigated.

### 2.2. Antibiotic Susceptibility

#### 2.2.1. LASARUS Agar Dilution

MIC profiles were available for forty-eight isolates, comprising sixteen of the tested *Legionella* species, after 96 h of incubation (Table 1). Inconsistent growth on LASARUS across biological replicates for five additional species (*L. cherrii*, *L. donaldsonii*, *L. dresdenensis*, *L. moravica*, and *L. nautarum*) was recorded but not included here due to incomplete MIC profiles (see Appendix A). Isolates have been grouped by species where ten or more representative isolates were available. For at least one species group (*L. pneumophila* (n = 13), *L. longbeachae* (n = 10), or other *Legionella* species (n = 25)), a significantly different median MIC was identified across all antibiotics, excluding levofloxacin (Figure 1).

Thirteen *Legionella* species had MIC values ± 3 dilutions from the mean MIC of *L. pneumophila* for all antibiotics, excluding levofloxacin. MICs > 3 dilutions higher than the mean MIC of *L. pneumophila* with ampicillin, azithromycin, or rifampicin were observed in *L. anisa*, *L. longbeachae*, *L. quinlivanii*, and *L. rubrilucens*. A > 3 dilutions lower MIC than the mean of *L. pneumophila* was observed in the species *L. anisa*, *L. dumoffii*, *L. feeleii*, *L. gormanii*, *L. israelensis*, *L. jordanis*, *L. londiniensis*, *L. longbeachae*, *L. micdadei*, *L. parisiensis*, *L. rubrilucens*, and *L. shakespearei* against ampicillin, chloramphenicol, or doxycycline. Table 2 outlines the MIC ranges, MIC_50_, and MIC_90_ concentrations of isolates within each of the species’ groups.

#### 2.2.2. Broth Microdilution

The growth of 33/34 tested *Legionella* spp. was supported to obtain complete MIC profiles for 83 isolates after 72 h utilising BYE-BMD (Table 1). The only species to not grow in the broth was *L. wadsworthii*. To compare directly with the incubation time requirements of LASARUS and BCYE, the results shown here for BYE-BMD were collected after 96 h of incubation (Figure 2). See Appendix A for the MICs obtained after 72 h. Isolates have been grouped by species where ten or more representative isolates were available. Across at least one of the species groups (*L. pneumophila* (n = 13), *L. longbeachae* (n = 10), *L. anisa* (n = 10), or other *Legionella* species (n = 50)), a significantly different median MIC was observed against ampicillin, azithromycin, doxycycline, and levofloxacin.

Twenty-six *Legionella* spp. had MIC values 3 dilutions lower than the mean MIC of *L. pneumophila* for doxycycline and rifampicin. A relative MIC difference was also seen for nine of these species (*L. anisa*, *L. beliardensis*, *L. cherrii*, *L. fallonii*, *L. jordanis*, *L. londiniensis*, *L. longbeachae*, *L. oakridgensis*, and *L. quinlivanii*) against azithromycin and two species (*L. dresdenensis* and *L. jordanis*) against ampicillin. Table 3 outlines the MIC ranges, MIC_50_, and MIC_90_ concentrations of isolates within each of the species groups.

#### 2.2.3. BCYE Agar Dilution

Only six isolates, representing *L. anisa* (n = 4), *L. sainthelensi* (n = 1), and *L. wadsworthii* (n = 1), were tested using BCYE agar dilution, having failed to complete susceptibility testing in both the LASARUS agar dilution and BYE-BMD. Three *L. pneumophila* strains were included for species comparisons. Figure 3 outlines the MIC distributions across all six tested antibiotics. For five of the six tested antibiotics, *L. anisa* had a greater range of MICs compared to *L. pneumophila*.

### 2.3. Differences Between Legionella Species

*Legionella* species, with more than five representative isolates, were compared using the proposed standardised BMD method of Sewell et al. (Figure 4) [21]. This included *L. anisa* (n = 10), *L. feeleii* (n = 6), *L. longbeachae* (n = 10), *L. pneumophila* (n = 13), and *L. rubrilucens* (n = 7). The overall comparisons of the five species using the Kruskal–Wallis test were significant for ampicillin (*p* ≤ 0.0001), azithromycin (*p* ≤ 0.001), doxycycline (*p* ≤ 0.001), levofloxacin (*p* ≤ 0.001), and rifampicin (*p* ≤ 0.01). Dunn’s multiple comparisons identified *L. feeleii* with ampicillin and doxycycline, *L. rubrilucens* with azithromycin and doxycycline, and *L. anisa* with levofloxacin to have significantly different susceptibilities compared to *L. pneumophila*.

### 2.4. Agreement of Legionella Antibiotic Susceptibility Testing Methods

The reproducibility of MICs obtained across LASARUS, BYE-BMD, and BCYE agar dilutions was evaluated by using three serogroup 1 *L. pneumophila* reference strains (NCTC 11192; NCTC 12008; NCTC 12024). Figure 5 summarises the MICs obtained for these three strains, across the three methods, for all tested antibiotics after 96 h of incubation. The results for BYE-BMD have been included for a 72 h incubation, as it was the earliest time point for completed MICs across the tested *Legionella* species. Significant differences in MICs obtained across the three methods were observed for ampicillin (*p* ≤ 0.01), chloramphenicol (*p* ≤ 0.05), and levofloxacin (*p* ≤ 0.01) using the Friedman test. The susceptibilities determined using LASARUS and BYE-BMD were not found to be significantly different.

Additionally, comparisons for the MICs of other *Legionella* species could be made for those successfully growing in broth and on LASARUS (Table 1). For all non-*pneumophila* species, the MICs obtained using BYE-BMD after 72- and 96 h and LASARUS after 96 h were found to be overall significantly different using the Friedman test (Table 4). For *L. longbeachae*, BYE-BMD MICs recorded after 96 h were found to be significantly higher than LASARUS after 96 h for ampicillin and azithromycin (Figure 6A). A significant increase in doxycycline MICs was observed between 72- and 96 h in BYE-BMD, and, for rifampicin BYE-BMD after 72 h, the MIC was significantly lower than in both LASARUS and BYE-BMD after 96 h. A greater lack of MIC reproducibility was observed in the mixed collection of *Legionella* species (Figure 6B).

## 3. Discussion

Official guidelines for standardised methods of antibiotic susceptibility testing, resistance thresholds, and ECOFF (Epidemiological Cut-Off) values for *Legionella* species remain undefined [22]. The designation of such criteria would provide a basis for identifying potential antibiotic resistance, or reduced susceptibility, emerging in *Legionella* spp. to clinically important antibiotics. Antibiotics which are specifically important for the treatment of *Legionella* infections are those which can act intracellularly and can accumulate at therapeutic concentrations within alveolar macrophages [23]. Currently, *Legionella* is widely regarded as susceptible to all antibiotics used to treat legionellosis [24]. However, the recent identification of *L. pneumophila* strains with a high macrolide MIC (erythromycin and azithromycin MICs ≥ 1024 mg/L) indicates that the few effective antibiotics for this species will need systematic surveillance [25,26].

Here, we report the antibiotic susceptibility data for thirty-three non-*pneumophila Legionella* species utilising a multi-site validated broth microdilution method for *L. pneumophila* [21]. Additionally, the susceptibility data for sixteen *Legionella* species have been presented through an implementation of the solid medium LASARUS [16]. Prior AST publications on *Legionella* species have utilised a variety of methods, including BYE-BMD, BCYE agar dilution, and E-test gradient strips on BCYE agar [19,27,28,29]. Historically, all *Legionella* required activated charcoal to absorb toxins for growth on solid media. Most of our tested isolates yielded a successful MIC determination utilising non-charcoal-based media, including LASARUS agar dilution (53.9%) or BYE-BMD (93.3%). However, six isolates required AST on BCYE agar for growth. The results collected from growth on charcoal-based media have been widely acknowledged to cause elevated MICs due to the chelating effect of activated charcoal [16,30,31]. As these samples were tested alongside *L. pneumophila* type strains, the MIC values, while not directly interpretable, showed no significant differences to *L. pneumophila*. Furthermore, some non-*pneumophila Legionella* species can require up to 2 weeks to achieve measurable growth [32]. Both LASARUS and the BYE-BMD protocols were refined using *L. pneumophila*, not other *Legionella* species. It is possible that longer incubation times may have generated more results, but the loss of antibiotic potency with longer incubation times is a significant concern to result validity.

Whilst *L. pneumophila* is regarded as the most clinically relevant *Legionella* species, the increased rate of *L. longbeachae*-related legionellosis in parts of the southern hemisphere, and increasing rates globally, emphasise the clinical importance of non-*pneumophila* species [6]. Isenman et al. evaluated the MICs of 61 clinical *L. longbeachae* against six antibiotics using both BYE-BMD and E-test gradient strips on BCYE [33]. To date, and as far as we are aware, no other study has exclusively published AST data focused on clinically sourced *L. longbeachae*. Other papers have published AST for *L. longbeachae*; however, they have incorporated only select type strains and environmentally sourced isolates, much like the work we have presented [34,35]. The ranges for azithromycin and rifampicin MICs have been reported as 0.032–0.25 mg/L, 0.06–0.25 mg/L, 0.06–0.5 mg/L and ≤0.008–0.064, ≤0.004–0.03, and ≤0.002–0.06, respectively, by Ienman et al., Gómez-Lus et al., and Nimmo and Bull [33,34,35]. Ienman et al. and Gómez-Lus et al. conducted AST with BYE-BMD, whereas Nimmo and Bull obtained MIC values using disc diffusion on BCYE agar. These ranges overlap with our ranges of 0.032–0.5 mg/L for azithromycin and 0.001–0.016 mg/L for rifampicin. Gómez-Lus et al. additionally reported MICs for levofloxacin, with a range of 0.008–0.016 mg/L, and an MIC_50_ and MIC_90_ of 0.016 mg/L. Our study found *L. longbeachae* to have a two-fold increase in MIC_50_ and MIC_90_ at 0.032 mg/L, and an elevated MIC range of 0.032–0.125 mg/L.

Levofloxacin MICs have been reported by Stout et al. for *L. pneumophila*, *L. micdadei*, *L. bozemanae*, and *L. jordanis*, obtained using BYE-BMD [36]. Our study identified respective MICs of 0.032 mg/L and 0.016 mg/L for clinically and environmentally sourced strains of *L. micdadei* and 0.016 mg/L for *L. bozemanae*, comparable to the 0.015 mg/L previously reported for each species. Bopp et al. have additionally reported an MIC of 0.015 mg/L for *L. micdadei*, utilising BYE macrotube dilution [37]. A two-fold higher MIC was observed by Stout et al. for *L. jordanis*, with an MIC of 0.06 mg/L, similar to our 0.032 mg/L. We observed variable levofloxacin MICs in *L. pneumophila*, ranging from 0.016 to 0.064 mg/L, correlating with prior published BYE-BMD ranges [24,34,36,38,39]. Gómez-Lus et al. have previously reported data on *L. dumoffii* and *L. gormanii* for eight antibiotics, including azithromycin, levofloxacin, and rifampicin [34]. The results collected in this study were within the published MIC ranges for azithromycin (0.125–0.5 mg/L) and rifampicin (<0.004–0.008 mg/L). However, for *L. dumoffii*, we identified MICs of 0.032 mg/L and for *L. gormanii* an MIC of 0.125 mg/L with levofloxacin, 2-fold and 8-fold higher than the respective maximum range Gómez-Lus et al. published.

Whilst we identified multiple, statistically significant differences between the MICs obtained in LASARUS agar dilution or BYE-BMD, as discussed above, our findings have aligned with previously published data. For our *L. pneumophila* control strains, ampicillin was the only antibiotic identified to differ significantly between the two methods, with MICs being notably higher (range ≥ 2 mg/L) in BMD compared to LASARUS agar dilution (range 0.25–1 mg/L) for this species. However, as our MIC range was capped at 2 mg/L, it is possible we were only observing a 2–4-fold increase in the maximum MIC against ampicillin in BMD. Our ampicillin MICs in LASARUS are comparable to those reported in broth by Wilson et al. (range 0.125–1 mg/L) [40]. Across the non-*pneumophila Legionella* species, there was a reduced concordance between the two methods. This may be due to the optimisation of both AST methodologies for *L. pneumophila* and highlights a need to re-evaluate the existing methods to support a greater testing of non-*pneumophila* species.

Despite the reduced concordance in non-*pneumophila* ASTs between methods, performing biological replicates alongside *L. pneumophila* type strains has provided an important basis for comparison. We have observed across clinically relevant non-*pneumophila Legionella* species that MICs can differ significantly from those of *L. pneumophila*. This includes the elevated MICs against azithromycin, the first-line macrolide therapeutic for the treatment of LD, of six *Legionella* species in LASARUS agar dilution (*L. erythra*, *L. gormanii*, *L. londiniensis*, *L. longbeachae*, *L. quinlivanii*, and *L. rubrilucens*) and nine in BYE-BMD (*L. bozemanae*, *L. cherrii*, *L. erythra*, *L. fairfieldensis*, *L. gormanii*, *L. maceachernii*, *L. micdadei*, *L. rubrilucens*, and *L. spiritensis*) which shared an MIC of >1 mg/L with one *L. pneumophila* strain. Such differences emphasise the importance of improved diagnostics for non-*pneumophila Legionella*, with microbial culture remaining the most accurate or available method, and the availability of MIC data [41,42,43].

Whilst there are currently no defined antibiotic resistance breakpoints for *Legionella*, guidance published by The European Committee on Antimicrobial Susceptibility Testing (EUCAST) has advised on methods for detecting antibiotic resistance mechanisms in *L. pneumophila* by comparing MIC data to defined ECOFF values [26]. This has been supported by the identification of antibiotic resistance genes in *L. pneumophila*, including *gyrA* (point mutation resulting in ciprofloxacin resistance) and *lpeAB* (efflux pump associated with macrolide resistance) [24,44]. Additionally, a resistance to fluoroquinolones has been reported in association with a *gyrA83* mutation [45]. Investigations into *L. pneumophila* have since attributed elevated MICs to these defined genes [46,47,48,49]. However, numerous studies have also reported elevated azithromycin MICs in samples negative for the presence of the *lpeAB* gene, highlighting the need for continued investigations into currently unidentified antibiotic resistance mechanisms within *Legionella* species [16,18,46,50].

Many research gaps remain in our understanding of emerging or undefined antibiotic resistance in *Legionella* species. This is particularly true for non-*pneumophila* species, which, as demonstrated in this study, have the potential to exhibit a reduced susceptibility to antibiotics that are otherwise effective in treating *Legionella* infections. The potential emergence of reduced macrolide or fluoroquinolone susceptibility in environmental or clinical *Legionella* strains poses a significant public health concern, particularly among vulnerable populations and during healthcare-associated outbreaks [51,52]. Strengthening AMR surveillance and establishing standardised AST protocols for *Legionella* are therefore critical for maintaining effective treatment options and informing global antimicrobial stewardship efforts [22].

This study has also highlighted the current limitations faced when trying to establish a standardised AST protocol for all *Legionella* species. All samples grew on BCYE; however, AST of all samples was not conducted on BCYE due to the previously mentioned limitations of the method. The successful growth of the whole isolate panel could not be obtained using LASARUS or BYE media, potentially relating to nutrient and charcoal differences compared to BCYE. An inconsistent growth was observed for five *Legionella* species on the LASARUS medium, and these were excluded to ensure data consistency and prevent misinterpretation, as incomplete or unreliable MIC profiles could bias comparative analyses. Consequently, MIC data for these five species were not presented in the figures. This represents a limitation of the study and underscores the challenges of achieving consistent growth conditions across diverse *Legionella* species. Despite this, the published BYE-BMD method obtained non-*pneumophila* MICs within 72 h, and *L. pneumophila* within 48 h, providing a successful AST alternative for thirty-three *Legionella* species faster than the tested agar-based methods [21]. It is standard practice to perform single determinations for measuring MICs in microbiology; we have performed all measurements as biological replicates (with triplicates where results differed by 2-fold dilution); however, as we were developing this as a comparative method, not performing all measurements in triplicate could be viewed as a limitation to this study.

Although LASARUS agar dilution performed less consistently, it remains advantageous for its high-throughput testing capacity with multipoint inoculation, improved safety compared to BCYE due to the translucency of the agar (negating the requirement of opening plate lids due to condensation), and subsequent potential for result collection using automated optical systems [16]. BYE-BMD, while effective, is limited by its labour-intensive set up and contamination checks, and BCYE by its charcoal chelation and opacity. Overall, our findings provide a foundation for future AST standardisation and comparative analyses across 34 *Legionella* species, addressing a key gap in the characterisation of *Legionella* antimicrobial susceptibility.

## 4. Materials and Methods

### 4.1. Study Design

A total of 89 *Legionella* isolates, comprising 34 *Legionella* species and 36 registered NCTC strains, were supplied by the UK Health Security Agency (UKHSA; London, UK) (Table A1 and Table A2). Samples were collected from clinical and environmental sources (1977–2021).

Biological replicates were performed to obtain an n = 2. If MICs were ±2 dilutions apart, a third replicate was performed to correct discrepancies and identify outliers. Initially, AST of all isolates was conducted using LASARUS agar dilution and BYE-BMD. Isolates with no MICs determined in either method were put forward for agar dilution using BCYE. Three *L. pneumophila* NCTC strains (NCTC 11192; NCTC 12008; NCTC 12024) were included for testing on BCYE.

### 4.2. Microbial Culturing

All isolates were cultured on BCYE media from glycerol bead archived (stored −80 °C on microbank beads (Technical Service Consultants, Heywood, UK)) and incubated at 36 °C (±1 °C) for 72 to 96 h in a humidified atmosphere. These conditions were replicated for the incubation of LASARUS and BCYE agar dilution plates. BMD plates were incubated on a shaking incubator (100 rpm) (Luckham R100 rotatest shaker, Burgess Hill, UK) at 37 °C for 96 h.

### 4.3. Antimicrobial Susceptibility Testing

AST was conducted against six antibiotics: ampicillin, doxycycline, levofloxacin, rifampicin (Sigma-Aldrich, Poole, UK), azithromycin (Aspire Pharma, Petersfield, UK), and chloramphenicol (Sigma-Aldrich, Poole, UK). Working stocks were prepared fresh at concentrations of 2560 mg/L, 80 mg/L, and 2.5 mg/L and final concentration ranges for AST between 0.0005 and 128 mg/L.

The solid culture medium LASARUS was prepared in-house based on the previously available formulation (Instant Test Ltd., Blackwood, UK). BCYE (Sigma–Aldrich, Poole, UK) was prepared according to the manufacturer’s instructions. Antimicrobial susceptibility testing (AST) was performed on both LASARUS and BCYE as previously described, with the adaptation of using a 1:10 dilution of a 1 McFarland suspension [16]. BYE-BMD was prepared and conducted as previously reported [21].

### 4.4. Statistical Analysis

Statistical analysis was performed using GraphPad Prism version 10.5.0. All datasets were assessed for normality using the Shapiro–Wilk test. As most datasets did not meet the assumptions of normality, non-parametric methods were used. The Kruskal–Wallis test was applied for comparisons between species (independent groups), and the Friedman test for comparisons between methods (repeated measures). Where appropriate, post hoc analysis was conducted using Dunn’s multiple comparisons test. Log_2_-transformed MIC data were also assessed, but transformation did not result in a normal distribution, and thus non-parametric methods remained appropriate. All graphs were additionally generated in GraphPad Prism. Calculations of MIC_50_ and MIC_90_, the concentrations at which 50% and 90% of the isolate panel growth is inhibited, respectively, along with MIC ranges, were performed independently.

### 4.5. Ethical Approval

Ethical approval was not required due to the anonymisation of patient details by UKHSA prior to the receival of isolates at Cardiff University. No ethical approval is required for NCTC strains.

## 5. Conclusions

While significant efforts to standardise AST internationally for *L. pneumophila* using BYE-BMD are currently underway, *L. pneumophila* is not the only cause of legionellosis, and antimicrobial resistance thresholds may not be directly applicable to other *Legionella* species. While BYE-BMD was able to establish consistent MICs for all species tested, 4/10 *L. anisa* isolates (an emerging pathogen of concern) would not grow on any medium other than BCYE; furthermore, longer incubations were often required for non-*pneumophila* species to achieve turbidity (BMD) or colony formation (agars), which impacts direct comparisons. Ampicillin cannot be used to treat legionellosis; however, a majority of *Legionella* species had MICs >2 mg/L in BMD, which is a generalised pK/pD therapeutic efficacy cut-off. Regarding clinically relevant antimicrobials, *L. pneumophila* had higher or equivalent MIC distributions relative to all other species except for rifampicin, where *L. longbeachae* was significantly higher, and there are a number of *Legionella* species isolates with higher MICs for azithromycin. Given that azithromycin is a first-line therapeutic, resistance thresholds may have to be established for each species individually in the future.

## Figures and Tables

**Figure 1 antibiotics-14-01165-f001:**
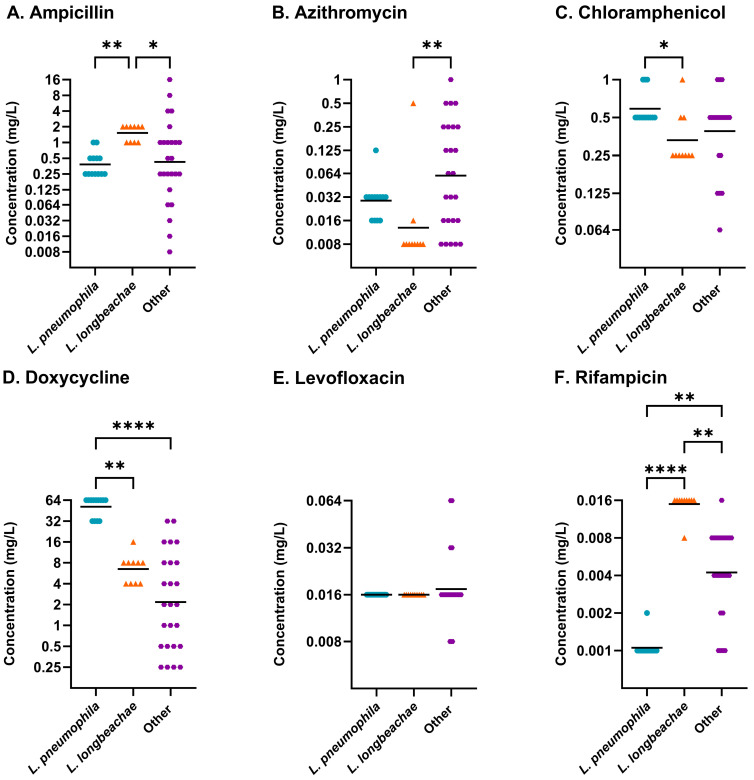
Distribution of antibiotic susceptibilities determined using LASARUS agar dilution at 96 h incubation for (**A**) ampicillin, (**B**) azithromycin, (**C**) chloramphenicol, (**D**) doxycycline, (**E**) levofloxacin, and (**F**) rifampicin. *L. pneumophila* (blue circle), *L. longbeachae* (orange triangle), and other *Legionella* species (purple hexagon) were identified to have overall significantly different MICs through utilisation of non-parametric Kruskal–Wallis analysis for five antibiotics (ampicillin *p* ≤ 0.01; azithromycin *p* ≤ 0.01; chloramphenicol *p* ≤ 0.05; doxycycline *p* ≤ 0.0001; rifampicin *p* ≤ 0.0001). The geometric mean (bar) and significant differences between species collections, determined using post hoc Dunn’s multiple comparisons test, are shown. MICs (mg/L) are shown on a Log_2_ scale to reflect doubling dilutions. (Abbreviations: *p* ≤ 0.05, *; *p* ≤ 0.01, **; *p* ≤ 0.0001, ****).

**Figure 2 antibiotics-14-01165-f002:**
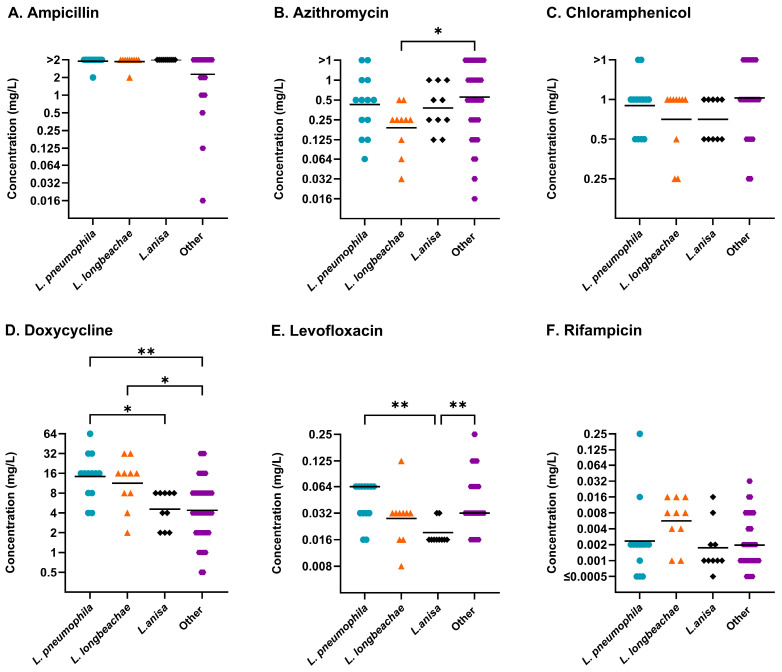
Distribution of antibiotic susceptibilities determined using BYE-BMD after 96 h shaking incubation for (**A**) ampicillin, (**B**) azithromycin, (**C**) chloramphenicol, (**D**) doxycycline, (**E**) levofloxacin, and (**F**) rifampicin. Individual *Legionella* species are shown separately where data was available for >10 isolates. *L. pneumophila* (blue circle), *L. longbeachae* (orange triangle), *L. anisa* (black diamond), and other *Legionella* species (purple hexagon) were identified to have overall significantly different MICs through utilisation of non-parametric Kruskal–Wallis analysis for four antibiotics (ampicillin *p* ≤ 0.01; azithromycin *p* ≤ 0.05; doxycycline *p* ≤ 0.001; levofloxacin *p* ≤ 0.01). The geometric mean (bar) and significant differences between species collections, determined using post hoc Dunn’s multiple comparisons test, are shown. MICs (mg/L) are shown on a Log_2_ scale to reflect doubling dilutions. (Abbreviations: *p* ≤ 0.05, *; *p* ≤ 0.01, **).

**Figure 3 antibiotics-14-01165-f003:**
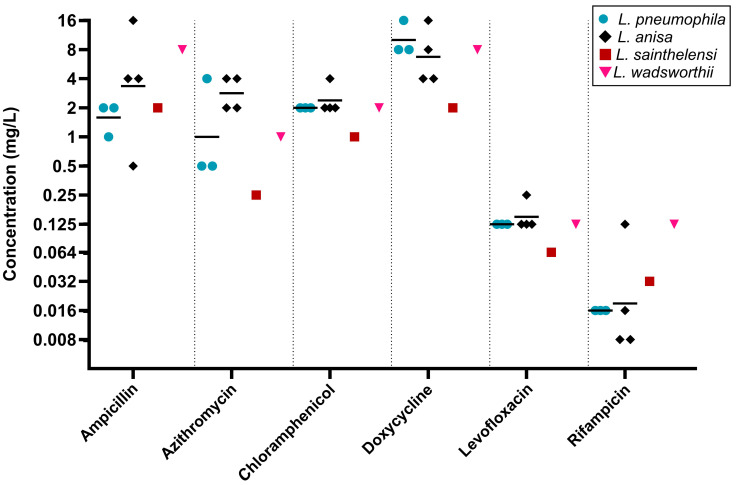
Distribution of antibiotic susceptibilities determined using BCYE agar dilution after 96 h. For *L. pneumophila* (blue circle) and *L. anisa* (black diamond), the geometric mean is shown (bar). Susceptibility ranges of *L. pneumophila*, *L. anisa*, and *L. wadsworthii* (pink inverted triangle) overlapped for all six antibiotics. *L. sainthelensi* (red square) demonstrated reduced susceptibility against azithromycin, chloramphenicol, doxycycline, and levofloxacin. MICs (mg/L) are shown on a Log_2_ scale to reflect doubling dilutions.

**Figure 4 antibiotics-14-01165-f004:**
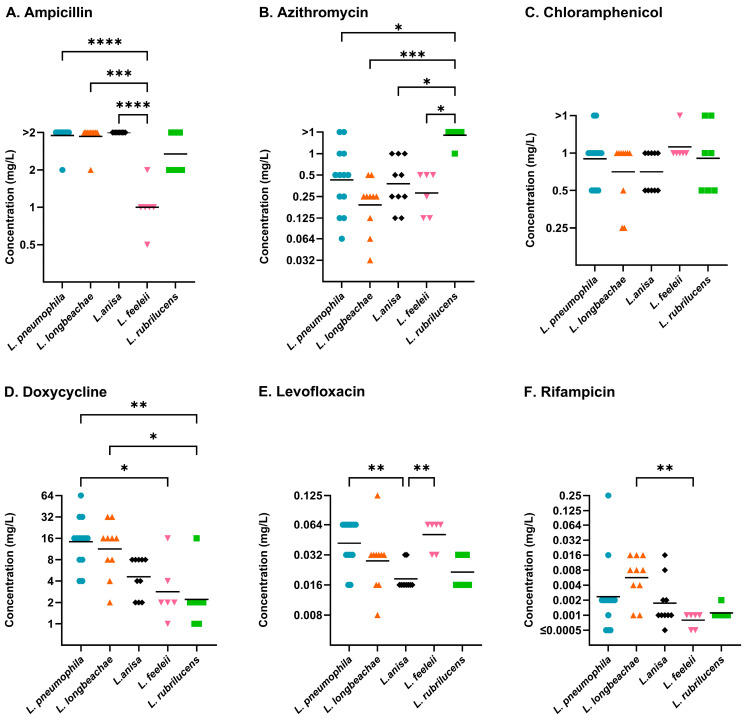
Comparative susceptibility of *L. anisa* (black diamond), *L. feeleii* (pink inverted triangle), *L. longbeachae* (orange triangle), and *L. rubrilucens* (green square) benchmarked against *L. pneumophila* (blue circle), using BYE-BMD for clinically relevant antibiotics (ampicillin (**A**), azithromycin (**B**), chloramphenicol (**C**), doxycycline (**D**), levofloxacin (**E**), and rifampicin (**F**)). The geometric mean (bar) and significant differences between species collections, determined using post hoc Dunn’s multiple comparisons test, are shown. MICs (mg/L) are shown on a Log_2_ scale to reflect doubling dilutions. (Abbreviations: *p* ≤ 0.05, *; *p* ≤ 0.01, **; *p* ≤ 0.001, ***; *p* ≤ 0.0001, ****).

**Figure 5 antibiotics-14-01165-f005:**
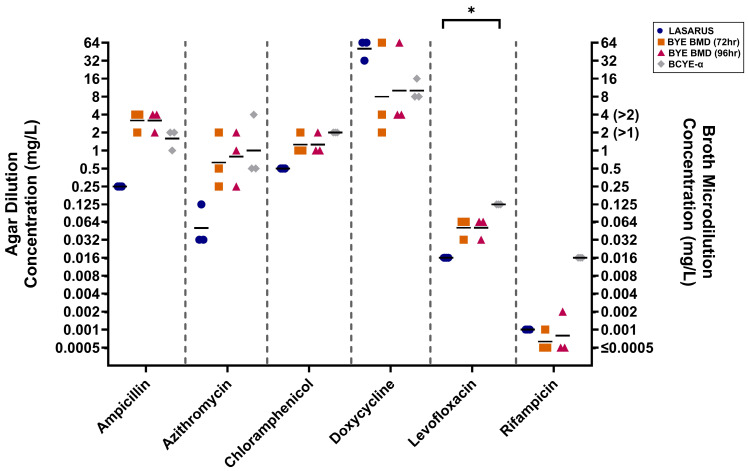
Three serogroup 1 *L. pneumophila* strains had MICs determined across LASARUS agar dilution (purple circle), BYE-BMD (pink triangle), and BCYE agar dilution (grey diamond) for six antibiotics after 96 h incubation. MICs obtained using BYE-BMD after 72 h (orange square) have been included as a comparison for the earliest read time. The geometric mean (bar) and significant differences between MICs for single antibiotics, identified using post hoc Dunn’s multiple comparisons test, are shown. Antibiotic ranges on the broth axis in brackets represent capped test ranges for azithromycin (>1 mg/L), chloramphenicol (>1 mg/L), and ampicillin (>2 mg/L). MICs (mg/L) are shown on a Log_2_ scale to reflect doubling dilutions. (Abbreviations: *p* ≤ 0.05, *).

**Figure 6 antibiotics-14-01165-f006:**
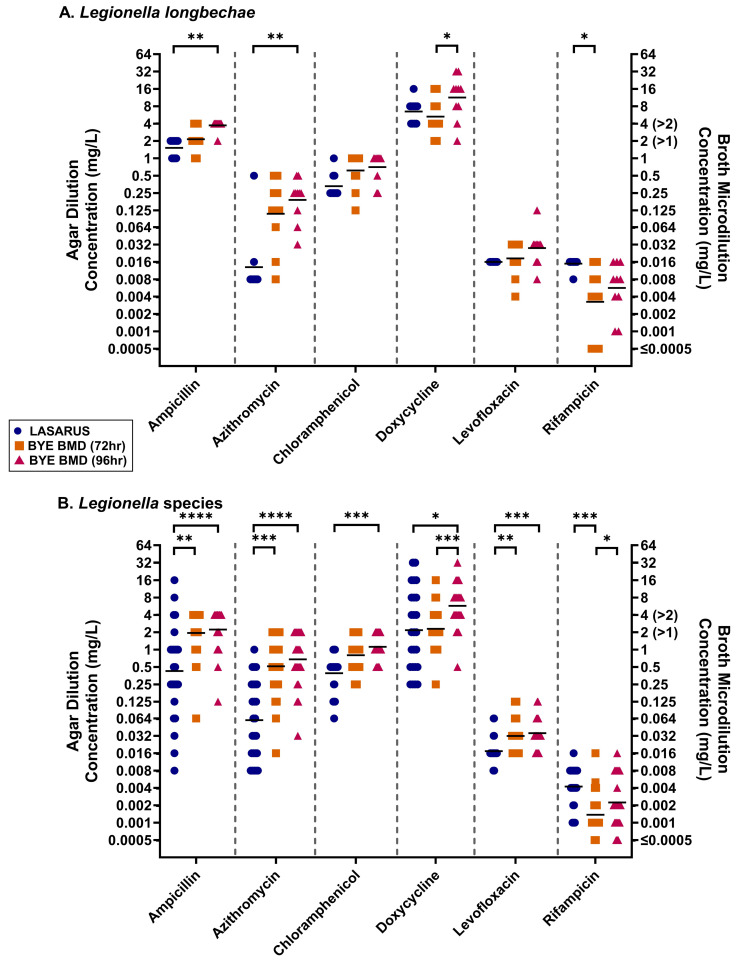
After 72 h incubation in BYE-BMD (orange square) and 96 h incubation in LASARUS agar dilution (blue circle) and BYE-BMD (pink triangle), MICs could be obtained for *L. longbeachae* (**A**) and a collection of other *Legionella* species (**B**). The geometric mean (bar) and significant differences between MICs obtained for single antibiotics, identified using post hoc Dunn’s multiple comparisons test, are shown. Antibiotic ranges on the broth axis in brackets represent capped test ranges for azithromycin (>1 mg/L), chloramphenicol (>1 mg/L), and ampicillin (>2 mg/L). MICs (mg/L) are shown on a Log_2_ scale to reflect doubling dilutions. (Abbreviations: *p* ≤ 0.05, *; *p* ≤ 0.01, **; *p* ≤ 0.001, ***; *p* ≤ 0.0001, ****).

**Table 1 antibiotics-14-01165-t001:** Method of *Legionella* species antibiotic susceptibility testing.

*Legionella* Species	Isolates (n)	Susceptibility Profiles ^1^ (n (%))		
		LASARUS	Broth	BCYE
*L. anisa*	14	1 (7.14)	10 (83.33)	4 (100)
*L. beliardensis*	1	-	1 (100)	n/a
*L. birminghamensis*	1	-	1 (100)	n/a
*L. bozemanae*	1	-	1 (100)	n/a
*L. brunensis*	1	-	1 (100)	n/a
*L. busanensis*	1	-	1 (100)	n/a
*L. cherrii*	3	-	3 (100)	n/a
*L. cincinnatiensis*	1	1 (100)	1 (100)	n/a
*L. donaldsonii*	1	-	1 (100)	n/a
*L. dresdenensis*	1	-	1 (100)	n/a
*L. dumoffii*	2	2 (100)	2 (100)	n/a
*L. erythra*	2	1 (50)	2 (100)	n/a
*L. fairfieldensis*	1	-	1 (100)	n/a
*L. fallonii*	2	-	1 (100)	n/a
*L. feeleii*	6	4 (66.67)	6 (100)	n/a
*L. gormanii*	1	1 (100)	1 (100)	n/a
*L. hackeliae*	2	-	2 (100)	n/a
*L. israelensis*	1	1 (100)	1 (100)	n/a
*L. jamestowniensis*	1	-	1 (100)	n/a
*L. jordanis*	1	1 (100)	1 (100)	n/a
*L. londiniensis*	2	2 (100)	2 (100)	n/a
*L. longbeachae*	10	10 (100)	10 (100)	n/a
*L. maceachernii*	1	-	1 (100)	n/a
*L. micdadei*	2	1 (50)	2 (100)	n/a
*L. nautarum*	1	-	1 (100)	n/a
*L. oakridgensis*	1	-	1 (100)	n/a
*L. parisiensis*	1	1 (100)	1 (100)	n/a
*L. pneumophila*	13	13 (100)	13 (100)	3 (100)
*L. quinlivanii*	2	1 (50)	2 (100)	n/a
*L. rubrilucens*	7	7 (100)	7 (100)	n/a
*L. sainthelensi*	2	-	1 (100)	1 (100)
*L. shakespearei*	1	1 (100)	1 (100)	n/a
*L. spiritensis*	1	-	1 (100)	n/a
*L. wadsworthii*	1	-	-	1 (100)
∑	89	48 (53.93)	83 (93.26)	9 (100) *

^1^ For each method and species, the number and percentage of isolates with completed minimum inhibitory concentration profiles has been shown. All *Legionella* species were tested using LASARUS agar dilution and BYE broth microdilution. BCYE agar dilution only included isolates with no successful growth in the other methods and three *L. pneumophila* reference strains for comparison (NCTC 11192; NCTC 12008; NCTC 12024). Isolates where no testing was performed on BCYE are shown as n/a. (* the percentage isolate growth is displayed for the number of isolates which were put forward for testing and not representative of the whole species collection).

**Table 2 antibiotics-14-01165-t002:** Antibiotic susceptibility of *Legionella* in LASARUS agar dilution.

Organism (n)	Antibiotic	MIC (mg/L)		
		MIC_50_	MIC_90_	Range
*L. pneumophila* (13)	Ampicillin	0.25	1	0.25–1
	Azithromycin	0.032	0.032	0.016–0.125
	Chloramphenicol	0.5	1	0.5–1
	Doxycycline	64	64	32–64
	Levofloxacin	0.016	0.016	0.016
	Rifampicin	0.001	0.001	0.001–0.002
*L. longbeachae* (10)	Ampicillin	2	2	1–2
	Azithromycin	0.008	0.016	0.008–0.5
	Chloramphenicol	0.25	0.5	0.25–1
	Doxycycline	8	8	4–16
	Levofloxacin	0.016	0.016	0.016
	Rifampicin	0.016	0.016	0.008–0.016
Other *Legionella* species (25)	Ampicillin	0.5	4	0.008–16
	Azithromycin	0.064	0.5	0.008–1
	Chloramphenicol	0.5	1	0.064–1
	Doxycycline	2	16	0.25–32
	Levofloxacin	0.016	0.032	0.008–0.064
	Rifampicin	0.004	0.008	0.001–0.016

Isolates have been grouped by species where ten or more representative isolates were available. For each of the six tested antibiotics, the range of MICs and concentrations of antibiotic required to inhibit growth of 50% (MIC_50_) and 90% (MIC_90_) of the isolate panel have been presented.

**Table 3 antibiotics-14-01165-t003:** Antibiotic susceptibility in BYE broth microdilution.

Organism (n)	Antibiotic	MIC (mg/L)		
		MIC_50_	MIC_90_	Range
*L. pneumophila* (13)	Ampicillin	>2	>2	2–>2
	Azithromycin	0.5	>1	0.064–>1
	Chloramphenicol	1	>1	0.5–>1
	Doxycycline	16	32	4–64
	Levofloxacin	0.064	0.064	0.016–0.064
	Rifampicin	0.002	0.016	≤0.0005–0.25
*L. longbeachae* (10)	Ampicillin	>2	>2	2–>2
	Azithromycin	0.25	0.5	0.032–0.5
	Chloramphenicol	1	1	0.25–1
	Doxycycline	16	32	2–32
	Levofloxacin	0.032	0.032	0.008–0.125
	Rifampicin	0.008	0.016	0.001–0.016
*L. anisa* (10)	Ampicillin	>2	>2	>2
	Azithromycin	0.25	1	0.125–1
	Chloramphenicol	0.5	1	0.5–1
	Doxycycline	4	8	2–8
	Levofloxacin	0.016	0.032	0.016–0.032
	Rifampicin	0.001	0.008	≤0.0005–0.016
Other *Legionella* species (50)	Ampicillin	>2	>2	0.016–>2
	Azithromycin	0.5	>1	0.016–>1
	Chloramphenicol	1	>1	0.25–>1
	Doxycycline	4	16	0.5–32
	Levofloxacin	0.032	0.125	0.016–0.25
	Rifampicin	0.002	0.008	≤0.0005–0.032

Isolates have been grouped by species where ten or more representative isolates were available. For each of the six tested antibiotics, the range of MICs and concentrations of antibiotic required to inhibit growth of 50% (MIC_50_) and 90% (MIC_90_) of the isolate panel have been presented.

**Table 4 antibiotics-14-01165-t004:** Non-*pneumophila* significant differences in MICs across methods.

*Legionella* Group	Antibiotic	Friedman Test	Dunn’s Multiple Comparisons Test	
*L. longbeachae*	Ampicillin	***	LASARUS—BYE-BMD 96 h	**
	Azithromycin	***	LASARUS—BYE-BMD 96 h	**
	Chloramphenicol	*		ns
	Doxycycline	*	BYE-BMD 72 h—BYE-BMD 96 h	*
	Levofloxacin	*		ns
	Rifampicin	**	LASARUS—BYE-BMD 72 h	*
*Legionella* spp.	Ampicillin	****	LASARUS—BYE-BMD 72 h	**
			LASARUS—BYE-BMD 96 h	****
	Azithromycin	****	LASARUS—BYE-BMD 72 h	***
			LASARUS—BYE-BMD 96 h	****
	Chloramphenicol	****	LASARUS—BYE-BMD 96 h	***
	Doxycycline	***	LASARUS—BYE-BMD 96 h	*
			BYE-BMD 72 h—BYE-BMD 96 h	***
	Levofloxacin	****	LASARUS—BYE-BMD 72 h	**
			LASARUS—BYE-BMD 96 h	***
	Rifampicin	****	LASARUS—BYE-BMD 72 h	***
			BYE-BMD 72 h—BYE-BMD 96 h	*

Collections of non-*pneumophila Legionella* species that had successful MIC determination in LASARUS agar dilution and BYE-BMD were compared. Significant differences identified using Friedman test and post hoc Dunn’s multiple comparisons test are shown. (Abbreviations: ns, not significant, *p* ≤ 0.05, *; *p* ≤ 0.01, **; *p* ≤ 0.001, ***; *p* ≤ 0.0001, ****).

## Data Availability

Data available on request from the authors.

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
