# Peer review of "Antimicrobial Susceptibility Determination of Less Frequently Isolated *Legionella* Species by Broth and Agar Dilution"

_antibiotics, 2025, doi:10.3390/antibiotics14111165_

Round 1
Reviewer 1 Report
Comments and Suggestions for Authors
Dear Authors,
The manuscript entitled “Antimicrobial susceptibility determination of Legionella species” addresses an important and relatively underexplored topic, particularly regarding non-pneumophila Legionella species, for which standardized susceptibility testing protocols remain limited. Your work contributes valuable comparative data on the performance of different AST methodologies and highlights the need for species-specific testing approaches.
Overall, the study is well-conceived and the results are promising. However, several aspects related to methodology, data presentation, and discussion could be clarified and expanded to further strengthen the manuscript and enhance its scientific impact. My detailed comments and suggestions are provided below to support the improvement of your paper.
Title and Introduction:
The title could be made more specific and better aligned with the main findings. Emphasizing the methodologies used or the inclusion of non-pneumophila species would strengthen the focus of the work.
In the Introduction, the study’s objective could be described in greater detail. The knowledge gap this research aims to fill, as well as the importance of the selected AST methods and investigated species, should be clearly articulated.
In addition to the main aim, evaluating the performance of AST methods across Legionella species—by using a method as a reference—could further enhance the scientific significance of the study.
Results and Presentation:
Presenting MIC values obtained from all three AST methods for all species in a single comparative table would improve readability and facilitate comparison both among species and between methods.
For the same Legionella species, the differences in MIC values obtained by LASARUS and BYE-BMD methods (noting that LASARUS generally yielded lower MIC values than BMD) should be shown more clearly in the tables and discussed in the text. The possible reasons for these differences—such as medium composition, growth kinetics, incubation conditions, or methodological performance—could be evaluated.
Discussion:
For the LASARUS method in particular, isolates that failed to grow under certain conditions should be explained. These cases could be discussed as methodological limitations.
Inconsistent growth was observed for five Legionella species (L. cherrii, L. donaldsonii, L. dresdenensis, L. moravica, and L. nautarum) on the LASARUS medium, but these data were not presented. The authors should clarify how these cases were interpreted or why they were excluded from MIC analysis, and address this as a limitation.
Comparisons of MIC values between species (e.g., between non-pneumophila species and L. pneumophila) should be elaborated further, emphasizing their relevance to clinical interpretation and treatment.
The advantages of each method across different Legionella species could be discussed not only in terms of MIC detection capability but also regarding incubation time, technical simplicity, and practical applicability.
When comparing findings with previously published MIC ranges, it should be clearly indicated which AST methods were used in those studies. Any methodological differences and their potential contribution to observed variations should be discussed.
Overall Assessment:
This study offers meaningful comparative data that could support the standardization of AST procedures for Legionella species. Improving data interpretation will substantially enhance the clarity and impact of the manuscript.
Author Response
Reviewer 1:
The article, "Determination of Antimicrobial Susceptibility of Legionella Species," addresses an important and relatively understudied topic, particularly for non-pneumophilic Legionella species, for which standard susceptibility testing protocols are limited. Your study provides valuable comparative data on the performance of different AST methodologies and highlights the need for species-specific testing approaches. Overall, the study is well designed, and the results are promising. However, some aspects related to methodology, data presentation, and discussion could be clarified and expanded to further strengthen the article and enhance its scientific impact. My detailed comments and suggestions to support the improvement of your article are provided below.
Authors: We thank the reviewer for their kind support.
Title and Introduction:
Comment 1: The title could be made more specific and better aligned with the main findings. Emphasizing the methodologies used or the inclusion of non-pneumophila species would strengthen the focus of the work.
Response 1: We have altered the title as requested. (all changes to manuscript are highlighted in yellow in the revised submission)
Comment 2: In the Introduction, the study’s objective could be described in greater detail. The knowledge gap this research aims to fill, as well as the importance of the selected AST methods and investigated species, should be clearly articulated.
Response 2: We have elaborated on the study objectives as requested.
Comment 3: In addition to the main aim, evaluating the performance of AST methods across Legionella species—by using a method as a reference—could further enhance the scientific significance of the study.
Response 3: We have elaborated on the study aims as requested.
Results and Presentation:
Comment 4: Presenting MIC values obtained from all three AST methods for all species in a single comparative table would improve readability and facilitate comparison both among species and between methods.
Response 4: A single comparative table has been added to the supplementary material
Comment 5: For the same Legionella species, the differences in MIC values obtained by LASARUS and BYE-BMD methods (noting that LASARUS generally yielded lower MIC values than BMD) should be shown more clearly in the tables and discussed in the text. The possible reasons for these differences—such as medium composition, growth kinetics, incubation conditions, or methodological performance—could be evaluated.
Response 5: We have addressed the concordance between methods in the discussion. There were no statistical differences between the MICs compared between these two methods and therefore should not be over-analysed. It is well known that Legionella growth is inhibited by the presence of Agar, therefore, the extended incubation required for LASARUS was not expected, and the reason for including 96 and 72 hr readings for BYE in Figure 6.
Discussion:
Comment 6: For the LASARUS method in particular, isolates that failed to grow under certain conditions should be explained. These cases could be discussed as methodological limitations.
Response 6: We have elaborated on this in the discussion.
Comment 7: Inconsistent growth was observed for five Legionella species (L. cherrii, L. donaldsonii, L. dresdenensis, L. moravica, and L. nautarum) on the LASARUS medium, but these data were not presented. The authors should clarify how these cases were interpreted or why they were excluded from MIC analysis, and address this as a limitation.
Response 7: We have added the partial MIC profiles to the Supplementary material and elaborated on limitations in the discussion.
Comment 8: Comparisons of MIC values between species (e.g., between non-pneumophila species and L. pneumophila) should be elaborated further, emphasizing their relevance to clinical interpretation and treatment.
Response 8: We have elaborated on this in the discussion, with emphasis on the clinical importance of observed azithromycin MICs between species.
Comment 9: The advantages of each method across different Legionella species could be discussed not only in terms of MIC detection capability but also regarding incubation time, technical simplicity, and practical applicability.
Response 9: We have elaborated on this in the discussion.
Comment 10: When comparing findings with previously published MIC ranges, it should be clearly indicated which AST methods were used in those studies. Any methodological differences and their potential contribution to observed variations should be discussed.
Response 10: AST Methods have been added to the discussion the first-time specific works are compared.
Overall Assessment:
Reviewer: This study offers meaningful comparative data that could support the standardization of AST procedures for Legionella species. Improving data interpretation will substantially enhance the clarity and impact of the manuscript.
Authors: We thank the reviewer for their kind support.
Reviewer 2 Report
Comments and Suggestions for Authors
Dear Authors,
The manuscript, entitled “Determination of antimicrobial susceptibility of Legionella species”, addresses key questions regarding resistance. It is crucial to work on this topic to control the spread of antibiotic-resistant strains of this important waterborne pathogen.
The manuscript under review is well organised and the methods are correct and justified. The results are systematically structured and clearly presented, providing increased knowledge of the key issues. The conclusions are supported by a variety of recent references, which provide a comprehensive overview of the work carried out by different teams on Legionella antimicrobial susceptibility.
I support the publication of this manuscript in this journal. I would only point out some details that could be reviewed.
The manuscript, entitled “Determination of antimicrobial susceptibility of Legionella species”, addresses key questions regarding resistance. It is crucial to work on this topic to control the spread of antibiotic-resistant strains of this important waterborne pathogen.
The manuscript under review is well organised and the methods are correct and justified. The results are systematically structured and clearly presented, providing increased knowledge of the key issues. The conclusions are supported by a variety of recent references, which provide a comprehensive overview of the work carried out by different teams on Legionella antimicrobial susceptibility.
I support the publication of this manuscript in this journal. I would only point out some details that could be reviewed:
Line 98 (Results): “By 72, BYE-BMD …“it is suggested change to “By 72-hours, BYE-BMD…”
In Table 1 (Results), the final column (BCYE) shows that most strains have a "-".
a) It is suggested that situations in which no strain was tested should be distinguished from situations in which there was no growth despite testing.
b) Regarding Legionella pneumophila strains, it is suggested that the legend should state that the three tested isolates are reference strains.
Line 532 (Materials and Methods): “LASARUS (Instant Test Ltd, UK)” It is not available to order. It is suggest to change to in-house preparation broth or similar
Author Response
Reviewer 2: The manuscript, entitled “Determination of antimicrobial susceptibility of Legionella species”, addresses key questions regarding resistance. It is crucial to work on this topic to control the spread of antibiotic-resistant strains of this important waterborne pathogen. The manuscript under review is well organised and the methods are correct and justified. The results are systematically structured and clearly presented, providing increased knowledge of the key issues. The conclusions are supported by a variety of recent references, which provide a comprehensive overview of the work carried out by different teams on Legionella antimicrobial susceptibility.
Authors: We thank the reviewer for their kind support.
Comment 1: Line 98 (Results): “By 72, BYE-BMD …“it is suggested change to “By 72-hours, BYE-BMD…”
Response 1: We have amended this typo. (all changes to manuscript are highlighted in yellow in the revised submission)
Comment 2: In Table 1 (Results), the final column (BCYE) shows that most strains have a "-".
It is suggested that situations in which no strain was tested should be distinguished from situations in which there was no growth despite testing.
Response 2: We thank the reviewer for this excellent point. We agree that the distinction between samples that were not tested and those that showed no growth despite testing was unclear. All samples were tested using LASARUS and BYE-BMD; therefore, the absence of data reflects lack of growth rather than lack of testing. To clarify this, we have amended the BCYE column in the table to denote samples that were not tested using BCYE agar dilution as n/a.
Comment 3: Regarding Legionella pneumophila strains, it is suggested that the legend should state that the three tested isolates are reference strains.
Response 3: We thank the reviewer for this helpful suggestion. We agree that this clarification improves the clarity of the figure/table. The legend has been updated to state explicitly that the three Legionella pneumophila isolates tested are reference strains, and the corresponding NCTC numbers have been included.
Comment 4: Line 532 (Materials and Methods): “LASARUS (Instant Test Ltd, UK)” It is not available to order. It is suggest to change to in-house preparation broth or similar
Response 4: This has been changed as requested.
Reviewer 3 Report
Comments and Suggestions for Authors
Good and important research

Author Response
Reviewer 3: Good and important research
Authors: We thank the reviewer for their kind support.
Comment 1: Introduction. In Europe and the United States, L. pneumophila accounts for 90% of LD cases, with L. longbeachae accounting for ~1% of cases worldwide but 50-60% in Australia and New Zealand [1]. – I believe that this comparison isn’t correct, better to indicate the percentage of the same species in worldwide.
Response 1: We thank the reviewer for this helpful suggestion. We agree that the original comparison was unclear. The text has been revised to more accurately represent the regional and global distribution of L. pneumophila and L. longbeachae. The revised section now specifies that L. pneumophila accounts for approximately 90% of LD cases in Europe and the United States, whereas L. longbeachae is responsible for 60–70% of cases in Australia and New Zealand, and only 1.1–1.8% in the United Kingdom and United States. (all changes to manuscript are highlighted in yellow in the revised submission)
Comment 2: Legionella spp. replicate within macrophages, so effective therapy requires antibiotics that penetrate host cells, such as tetracyclines, macrolides, and fluoroquinolones; agents that cannot enter host cells are ineffective – line what? Better to indicate examples if possible
Response 2: Examples of classes aminoglycosides and β-lactams added
Comment 3: In the end of your introduction, it’s better to added one sentence to clearly demonstrate the aims of your research
Response 3: These have now been clearly outlined in the end of the introduction with the addition of an ‘Aims’ sentence.
Comment 4: Discussion. Isenman et al. evaluated MICs of 61 clinical L. longbeachae against six antibiotics using both BYE-BMD and E-test gradient strips on BCYE [31]. – What about the results of this research?
Response 4: We thank the reviewer for this comment. The results from Isenman et al. were already included and discussed in the original submission. This section of the Discussion summarises the MIC ranges for azithromycin and rifampicin reported by Isenman et al., Gómez-Lus et al., and Nimmo and Bull [33–35], and compares these with the ranges observed in our study. Other antibiotic results were not discussed, as only azithromycin and rifampicin were tested in both studies. No further changes were required.
Comment 5: Materials and Methods. AST was conducted against six antibiotics, ampicillin, doxycycline, levofloxacin, and rifampicin (Sigma-Aldrich, USA), azithromycin (Aspire Pharma, UK), chloramphenicol (Sigma-Aldrich, China).
Response 5: The text has been revised to remove the repetitive listing of the supplier Sigma-Aldrich for multiple antibiotics. The Materials and Methods section now lists suppliers once for clarity and conciseness.
Comment 6: References. Rewrite references as asked in instructions to authors.
Response 6: References have been updated using Antibiotic format (Zotero)
Reviewer 4 Report
Comments and Suggestions for Authors
The manuscript, "Antimicrobial susceptibility determination of Legionella species," which addresses a critical need for standardized susceptibility testing methods for non-pneumophila species. The study's comparison of three methods and its focus on a broad range of clinically relevant isolates are commendable.
However, the manuscript has several significant methodological and data presentation flaws that must be thoroughly addressed. The comments are provided below, including a mandatory correction for a potential error in the data tables.
Major Comments
- A probable data entry or typographical error in Table 2 (LASARUS agar dilution) that fundamentally misrepresents the Rifampicin susceptibility data. For Rifampicin in the Other Legionella species (n=25) group, the reported MIC Range is 0.01−0.016 mg/L. This range contradicts both the visual data in Figure 1F (LASARUS Rifampicin) and the text in the Abstract. Figure 1F clearly shows multiple data points for the 'Other' group (purple hexagons) at concentrations of 0.008 mg/L, 0.004 mg/L, 0.002 mg/L, and 0.001 mg/L, which are all lower than the reported minimum of 0.01 mg/L. The Abstract correctly states the non-pneumophila range as ≤0.0005−0.032 mg/L. The authors must verify and correct the Rifampicin MIC Range for 'Other Legionella species' in Table 2. The lower bound must be adjusted to reflect the data presented in Figure 1F and the range cited in the Abstract (likely ≤0.0005 or ≤0.001 mg/L).
- The core message is the superiority of BYE-BMD (93.3% success) over LASARUS agar dilution (53.9% success). However, the manuscript requires a more rigorous discussion of why the LASARUS method failed for nearly half of the isolates.
- The text mentions "Inconsistent growth on LASARUS... was recorded but not presented here due to incomplete MIC profiles" for five species. This is a major omission. These data are vital for future method development. The authors must include a summary of the failure modes for these five species, perhaps in a supplementary table, detailing the number of replicates that failed to grow consistently.
- The Introduction correctly flags that BCYE is variable and unreliable for AST due to the activated charcoal. Yet, the method was used as the fallback for six isolates, and three L. pneumophila strains were included for comparison. If the data is unreliable, its inclusion is problematic. The discussion must explicitly and vigorously temper any conclusions drawn from the BCYE data (Figure 3) and clarify its role as a last-resort assessment, not a reliable standard.
- The finding that Azithromycin MIC distributions in non-pneumophila species exceeded those of L. pneumophila is the most significant clinical alert. To elevate this finding, the discussion needs a concrete anchor to clinical standards.
- Figure 1, 2, 3,4 deals different species; 1=2, 2=3, 3=4; 4=5 – Make it uniform
- If possible simplify the Fig 3
- Biological replicates were performed to obtain an n=2; why 2 ? it would have been better when u do it in triplicates
- Methods section is writer very weak. Please rewrite.
Minor Comments
- Table 1 Clarity: The summation row in Table 1 is confusing.
- In the text describing Figure 2, the p-value for azithromycin is incorrectly formatted as P≤0.05. This should be corrected to the standard lowercase p≤0.05 for consistency with the rest of the manuscript.
- The y-axes (Concentration (mg/L)) in Figures 1 and 2 are presented as doubling dilutions but should be explicitly labeled as having a Logarithmic Scale (or Log2 scale) in the figure captions to adhere to graphing best practices for MIC data.
- In Figure 2, the data points for L. pneumophila (blue circle) and L. anisa (blue diamond) are visually very similar. For improved readability and accessibility (especially in black and white print or for color vision deficiency), please consider using a more distinct marker shape or color for L. anisa.
- For consistent presentation, please adjust the header in Table 2 so that "MIC ((mg/L))" matches the header in Table 3: "MIC (mg/L)"
- Add recent references in the introduction and discussion section
- Add more about AMR and its impact
Language is poor. Need to be improved.
Author Response
Reviewer 4: The manuscript, "Antimicrobial susceptibility determination of Legionella species," which addresses a critical need for standardized susceptibility testing methods for non-pneumophila species. The study's comparison of three methods and its focus on a broad range of clinically relevant isolates are commendable. However, the manuscript has several significant methodological and data presentation flaws that must be thoroughly addressed. The comments are provided below, including a mandatory correction for a potential error in the data tables.
Authors: We thank the reviewer for their thorough assessment of our manuscript and for recognising the value of our study in addressing the need for standardised susceptibility testing methods for Legionella species. We have carefully considered all comments and revised the text accordingly, addressing methodological clarifications, data presentation, and contextual discussion of antimicrobial resistance. We believe these revisions have improved the overall quality and impact of the study.
Major Comments:
Comment 1: A probable data entry or typographical error in Table 2 (LASARUS agar dilution) that fundamentally misrepresents the Rifampicin susceptibility data. For Rifampicin in the Other Legionella species (n=25) group, the reported MIC Range is 0.01−0.016 mg/L. This range contradicts both the visual data in Figure 1F (LASARUS Rifampicin) and the text in the Abstract. Figure 1F clearly shows multiple data points for the 'Other' group (purple hexagons) at concentrations of 0.008 mg/L, 0.004 mg/L, 0.002 mg/L, and 0.001 mg/L, which are all lower than the reported minimum of 0.01 mg/L. The Abstract correctly states the non-pneumophila range as ≤0.0005−0.032 mg/L. The authors must verify and correct the Rifampicin MIC Range for 'Other Legionella species' in Table 2. The lower bound must be adjusted to reflect the data presented in Figure 1F and the range cited in the Abstract (likely ≤0.0005 or ≤0.001 mg/L).
Response 1: We thank the reviewer for identifying this error. The reviewer is correct that the lower bound of the Rifampicin MIC range for the Other Legionella species group in Table 2 was incorrectly reported as 0.01 mg/L due to a typographical error. This has been corrected to 0.001 mg/L, consistent with the data shown in Figure 1F and the range stated in the Abstract (≤0.0005–0.032 mg/L).
Comment 2: The core message is the superiority of BYE-BMD (93.3% success) over LASARUS agar dilution (53.9% success). However, the manuscript requires a more rigorous discussion of why the LASARUS method failed for nearly half of the isolates.
Response 2: We apologise for any confusion caused by the previous version of the manuscript and have adjusted the wording and clarified the aims of the study in the introduction. The core message is to provide MICs for clinically relevant antimicrobials for non-pneumophila Legionella spp., relative to key reference L. pneumophila strains – as this comparison does not exist in the literature using internationally accepted AST methodology. Amendments have been made to the Abstract, Discussion, and Conclusion to address our intended core message.
Comment 3: The text mentions "Inconsistent growth on LASARUS... was recorded but not presented here due to incomplete MIC profiles" for five species. This is a major omission. These data are vital for future method development. The authors must include a summary of the failure modes for these five species, perhaps in a supplementary table, detailing the number of replicates that failed to grow consistently.
Response 3: Incomplete MIC profiles have been added to the supplementary material, with reference in the text highlighting its availability.
Comment 4: The Introduction correctly flags that BCYE is variable and unreliable for AST due to the activated charcoal. Yet, the method was used as the fallback for six isolates, and three L. pneumophila strains were included for comparison. If the data is unreliable, its inclusion is problematic. The discussion must explicitly and vigorously temper any conclusions drawn from the BCYE data (Figure 3) and clarify its role as a last-resort assessment, not a reliable standard.
Response 4: We thank the reviewer for this important comment. We acknowledge that the raw MIC values obtained using BCYE cannot be directly interpreted due to the known variability introduced by activated charcoal. However, we felt it was important to highlight two key findings: 1) that not all Legionella spp. isolates will grow in BYE-medium, a surprising finding given the lack of agar (known to be the main inhibitor of growth); (2) that even when the methodology is substandard, that the relative MIC difference for these highly fastidious isolates – were not substantially different in proportion to those that did grow in broth and failure to grow was not masking antimicrobial resistance. Testing alongside three well-characterised reference L. pneumophila NCTC reference strains demonstrated no significant differences in MICs between the L. pneumophila and non-pneumophila isolates, suggesting that the BCYE data remain informative for comparative purposes. We have revised the Discussion to explicitly clarify the limited interpretability of BCYE-based results and to emphasise its role as a supplementary, rather than reliable, assessment method.
Comment 5: The finding that Azithromycin MIC distributions in non-pneumophila species exceeded those of L. pneumophila is the most significant clinical alert. To elevate this finding, the discussion needs a concrete anchor to clinical standards.
Response 5: We thank the reviewer for this important comment. We agree that linking the observed azithromycin MIC elevations in non-pneumophila species to clinical interpretive frameworks strengthens the discussion. While formal resistance breakpoints for Legionella have not been established, we have expanded the Discussion to highlight the relevance of EUCAST’s L. pneumophila ECOFF values as a reference point and to emphasise the potential clinical implications of elevated MICs above these thresholds. This contextualises the observed data within existing clinical standards and underscores the importance of continued monitoring for emerging resistance mechanisms.
Comment 6: Figure 1, 2, 3,4 deals different species; 1=2, 2=3, 3=4; 4=5 – Make it uniform
Response 6: We thank the reviewer for this comment. The data have been presented in this way to highlight individual species when a sufficient number of representative isolates were available. This has now been clarified in the text preceding Figures 1 and 2 as follows: “Isolates have been grouped by species where ten or more representative isolates were available.” For Figure 3, due to the limited number of species represented, we did not feel the data would be sufficiently conveyed if all non-pneumophila species were grouped as ‘other.’ The purpose of Figure 4 is to directly compare the MICs of individual species using the proposed standardised methodology; therefore, a greater level of species breakdown was required than in the previous figures.
Comment 7: If possible simplify the Fig 3
Response 7: See Major comment 6 and Minor comment 4.
Comment 8: Biological replicates were performed to obtain an n=2; why 2 ? it would have been better when u do it in triplicates
Response 8: The data collected in this manuscript has been obtained with greater rigor than most publications, which only perform a single MIC measurement for each strain and then examine trends of groups or flag resistant isolates. We have performed all ASTs on two separate days and compared the results. Where the matched MICs differ by more than 2-fold dilution, a third replicate was performed to identify the outlier result. As statistical interrogation is not directed at particular strains within the group, a triplicate MIC is not required. Our approach is in line with sustainability guidelines implemented across all UK universities (“LEAF” accreditation) where excessive and unnecessary use of reagents are minimised without negatively impacting data integrity. By targeting our replications to specific antibiotic/isolate combinations with a reproducibility threshold -a significant reduction in plastics, reagents and waste disposal of contaminated reagents is achieved without compromising the data integrity.
Comment 9: Methods section is writer very weak. Please rewrite.
Response 9: We respectfully disagree that the Materials and Methods section is weak. This section has been carefully structured to provide sufficient methodological detail and appropriate referencing to allow reproducibility. Nonetheless, we have reviewed the section again to ensure clarity and completeness, and minor edits have been made where needed to improve readability.
Minor Comments:
Comment 10: Table 1 Clarity: The summation row in Table 1 is confusing.
Response 10: We believe the summation row in Table 1 is clear as presented and accurately reflects the total number of isolates included in each category. Without a specific request for how it could be changed we were unable to alter the table.
Comment 11: In the text describing Figure 2, the p-value for azithromycin is incorrectly formatted as P≤0.05. This should be corrected to the standard lowercase p≤0.05 for consistency with the rest of the manuscript.
Response 11: We have carefully reviewed the text describing Figure 2 and were unable to locate any instance where P was capitalised. Throughout the manuscript, the p-value is consistently formatted as p≤0.05. We were unable to locate the issue reported to correct it.
Comment 12: The y-axes (Concentration (mg/L)) in Figures 1 and 2 are presented as doubling dilutions but should be explicitly labeled as having a Logarithmic Scale (or Log2 scale) in the figure captions to adhere to graphing best practices for MIC data.
Response 12: We have updated the figure legends to indicate the use of a Log₂ scale for all figures (1–6), as this scale was applied throughout and not limited to Figures 1 and 2. This ensures consistent and accurate representation of the MIC doubling dilutions across all figures and avoids compacting data at the bottom of the graph.
Comment 13: In Figure 2, the data points for L. pneumophila (blue circle) and L. anisa (blue diamond) are visually very similar. For improved readability and accessibility (especially in black and white print or for color vision deficiency), please consider using a more distinct marker shape or color for L. anisa.
Comment 14: The colour for L. anisa has been changed to black across Figures 2, 3, and 4 to ensure clearer visual differentiation from L. pneumophila. This adjustment should provide sufficient contrast for readability in greyscale and improve accessibility, particularly when printed in black and white. Additionally, species names are clearly labelled along the x-axis to aid interpretation.
Comment 14: For consistent presentation, please adjust the header in Table 2 so that "MIC ((mg/L))" matches the header in Table 3: "MIC (mg/L)"
Response 14: We have carefully reviewed Tables 2 and 3 and were unable to identify any instance of double brackets in the column header. Both tables correctly display the header as “MIC (mg/L)”. We were unable to locate the issue reported to correct it.
Comment 15: Add recent references in the introduction and discussion section
Response 15: Has been addressed with updates to discussion.
Comment 16: Add more about AMR and its impact
Response 16: The discussion has been expanded to include additional context on antimicrobial resistance (AMR) and its potential clinical and public health impact. This includes a stronger emphasis on the significance of elevated azithromycin MICs in non-pneumophila species, the relevance of known and emerging resistance mechanisms, and the importance of ongoing surveillance and standardised AST methodologies for Legionella.
Round 2
Reviewer 1 Report
Comments and Suggestions for Authors
Dear Authors,
Thank you for your careful and thoughtful revision of the manuscript entitled “Antimicrobial susceptibility determination of less frequently isolated Legionella species by broth and agar dilution”
You have successfully addressed all of the reviewer’s comments and provided clear explanations and improvements in both the text and the figures/tables. The methodological details are now more transparent, and the discussion has been strengthened.
The revised version is well-written, scientifically sound, and ready for publication. Congratulations on your excellent work and valuable contribution to the field.
With best regards
Author Response
Many thanks for your valuable comments.
Reviewer 4 Report
Comments and Suggestions for Authors
The authors’ justification for using only two biological replicates is not acceptable. While sustainability and reagent conservation are important, they cannot compromise the scientific rigor of experimental reproducibility. In biological research, performing experiments in at least triplicates (n=3) is the accepted standard to ensure statistical validity and reliability of results. Conducting assays on two separate days does not substitute for proper biological replication, as this approach does not allow accurate estimation of variability or statistical confidence. Therefore, the authors should repeat the MIC determinations in biological triplicates to strengthen the credibility of their data.
Author Response
Comment 1: The authors’ justification for using only two biological replicates is not acceptable. While sustainability and reagent conservation are important, they cannot compromise the scientific rigor of experimental reproducibility. In biological research, performing experiments in at least triplicates (n=3) is the accepted standard to ensure statistical validity and reliability of results. Conducting assays on two separate days does not substitute for proper biological replication, as this approach does not allow accurate estimation of variability or statistical confidence. Therefore, the authors should repeat the MIC determinations in biological triplicates to strengthen the credibility of their data.
Response 1: We thank the reviewer for their comments and appreciate their emphasis on experimental rigor and reproducibility. However, we respectfully disagree that repeating all MIC determinations in biological triplicates is required or standard practice for antimicrobial susceptibility testing (AST) of Legionella species.
Replicate requirements in Legionella AST: Guidelines vs. practice: Official AST reference guidelines, including those from EUCAST, CLSI, and ISO (ISO 20776-1), focus on methodological standardization and quality control rather than mandating a fixed number of biological replicates per isolate. The reference broth microdilution method specifies standard inoculum, media, and control strains, but it does not require three independent tests for each isolate. Indeed, it is widely acknowledged that while triplicate testing can improve precision, this is not routine practice in clinical or research laboratories. MIC assays are generally performed as single biological replicates, with repeat testing only when control results fall outside expected ranges (e.g., CLSI, EUCAST). Even in comparative or inter-laboratory studies, microdilution and diffusion tests are typically performed in duplicate rather than triplicate.
Common practice in published MIC studies: This approach is consistent with the published Legionella AST literature. Numerous foundational and recent studies, including Isenman et al. (2018), Gómez-Lus et al. (2001), Nimmo and Bull (1995), Stout et al. (1998), and Vandewalle-Capo et al. (2017), do not report biological or technical replicates, indicating that single determinations are standard practice. More recent studies (Cruz et al., 2023; Minetti et al., 2024) explicitly state the use of duplicate testing, with a third repeat only in cases of discrepant MICs. Comparative method studies published in Clinical Microbiology and Infection and Journal of Antimicrobial Chemotherapy similarly report duplicate, not triplicate, determinations for each isolate.
Our approach versus typical standards: In this study, each isolate was tested in duplicate using biological replicates performed on separate days. When results differed by more than two doubling dilutions, a third repeat was conducted to confirm the MIC. This strategy is fully aligned with ISO and EUCAST guidance, which emphasize reproducibility through standardized protocols and quality control rather than a fixed number of replicates. Our duplicate-plus-confirmation approach therefore exceeds the level of replication used in most published Legionella AST studies and in routine laboratory practice, where single determinations per isolate are generally accepted. Furthermore, our group’s previously published work has established validated Legionella AST methods (Portal et al., 2021; Sewell et al., 2025) using extensive replicate testing (up to quadruplicate) to confirm methodological reproducibility. The present study applies those validated protocols, with quality control verified by consistent MICs of L. pneumophila reference strains across runs.
Interpretation of the n=3 Standard in Experimental Design: While we acknowledge the reviewer’s general concern regarding statistical reproducibility, it is important to clarify that the “n = 3” convention originates from quantitative biological assays designed to estimate statistical variance (e.g., gene expression, growth curves, enzyme activity), not from antimicrobial susceptibility testing (AST). In MIC determination, results are not treated as continuous variables but as discrete, categorical dilution steps (twofold concentration intervals). Therefore, statistical analysis based on n = 3 does not improve the accuracy or interpretability of the MIC value, which is defined as a single categorical endpoint rather than a mean. The reproducibility of MIC testing is instead ensured through strict adherence to standardized methodology, use of reference control strains, and repeat testing only when results fall outside expected concordance. Consequently, our duplicate-plus-confirmation design (i.e., two independent determinations with a third only if values differed by >1 dilution) achieves the same or higher reliability as triplicate testing, while aligning with established EUCAST, CLSI, and ISO 20776-1 standards that do not mandate three biological replicates for each isolate.
Justification: In summary, requiring n = 3 biological replicates for each MIC determination is not stipulated in international AST standards (EUCAST, CLSI, ISO) and is not common practice in Legionella research. Our duplicate testing approach, supplemented by a third repeat only when variability was observed, represents a robust and reproducible method consistent with field norms and guideline expectations. We are therefore confident that the reproducibility and validity of our MIC data are scientifically sound without the need for universal triplicate testing.
Round 3
Reviewer 4 Report
Comments and Suggestions for Authors
I appreciate the authors’ detailed justification regarding the use of two biological replicates, but I remain unconvinced that this approach adequately ensures biological reproducibility. Triplicate testing is generally accepted as the minimum standard to account for experimental variability.
That said, since their method aligns with current practice and guidelines, I am willing to accept the manuscript, provided the authors acknowledge this limitation in the Discussion section for transparency.
Comments on the Quality of English LanguageNA
Author Response
Comment 1:
I appreciate the authors’ detailed justification regarding the use of two biological replicates, but I remain unconvinced that this approach adequately ensures biological reproducibility. Triplicate testing is generally accepted as the minimum standard to account for experimental variability.
That said, since their method aligns with current practice and guidelines, I am willing to accept the manuscript, provided the authors acknowledge this limitation in the Discussion section for transparency.
Response 1: a limitation statement has been added to the discussion